# Shelf Life of Minced Pork in Vacuum-Adsorbed Carvacrol@Natural Zeolite Nanohybrids and Poly-Lactic Acid/Triethyl Citrate/Carvacrol@Natural Zeolite Self-Healable Active Packaging Films

**DOI:** 10.3390/antiox13070776

**Published:** 2024-06-27

**Authors:** Vassilios K. Karabagias, Aris E. Giannakas, Nikolaos D. Andritsos, Areti A. Leontiou, Dimitrios Moschovas, Andreas Karydis-Messinis, Apostolos Avgeropoulos, Nikolaos E. Zafeiropoulos, Charalampos Proestos, Constantinos E. Salmas

**Affiliations:** 1Department of Food Science and Technology, University of Patras, 30100 Agrinio, Greece; vkarampagias@upatras.gr (V.K.K.); nandritsos@upatras.gr (N.D.A.); aleontiu@upatras.gr (A.A.L.); 2Department of Material Science and Engineering, University of Ioannina, 45110 Ioannina, Greece; dmoschov@uoi.gr (D.M.); karydis.and@gmail.com (A.K.-M.); aavger@uoi.gr (A.A.); nzafirop@uoi.gr (N.E.Z.); 3Laboratory of Food Chemistry, Department of Chemistry, National and Kapodistrian University of Athens Zografou, 15771 Athens, Greece; harpro@chem.uoa.gr

**Keywords:** poly-lactic acid, triethyl citrate, carvacrol, natural zeolite, self-healable, active packaging, control release, desorption kinetics, minced pork, shelf life

## Abstract

Enhancing food preservation and safety using environmentally friendly techniques is urgently needed. The aim of this study was to develop food packaging films using biodegradable poly-L-lactic acid (PLA) as biopolymer and carvacrol (CV) essential oil as an antioxidant/antibacterial agent for the replacement of chemical additives. CV was adsorbed onto natural zeolite (NZ) via a new vacuum adsorption method. The novel nanohybrid CV@NZ with a high CV content contained 61.7%wt. CV. Pure NZ and the CV@NZ nanohybrid were successfully dispersed in a PLA/triethyl citrate (TEC) matrix via a melt extrusion process to obtain PLA/TEC/xCV@NZ and PLA/TEC/xNZ nanocomposite films with 5, 10, and 15%wt CV@NZ or pure NZ content. The optimum resulting film PLA/TEC/10CV@NZ contained 10%wt. CV@NZ and exhibited self-healable properties, 22% higher tensile strength, 40% higher elongation at break, 45% higher water barrier, and 40% higher oxygen barrier than the pure PLA/TEC matrix. This film also had a high CV release content, high CV control release rate as well as 2.15 mg/L half maximal effective concentration (EC50) and 0.27 mm and 0.16 mm inhibition zones against *Staphylococcus aureus* and *Salmonella enterica* ssp. *enterica* serovar Typhimurium, respectively. This film not only succeeded in extending the shelf life of fresh minced pork, as shown by the total viable count measurements in four days but also prevented the lipid oxidation of fresh minced pork and provided higher nutritional values of the minced meat, as revealed by the heme iron content determination. It also had much better and acceptable sensory characteristics than the commercial packaging paper.

## 1. Introduction 

The goal of reducing global emissions sufficiently to meet global climate goals and avoid severe and catastrophic impacts of climate change is becoming increasingly important [1,2]. This makes the necessity for the replacement of plastics derived from fossil fuels imperative [3,4]. Plastics can be replaced by natural biodegradable biopolymers such as cellulose, starch, and chitosan or by biodegradable polymers, which can be developed from waste and byproducts such as poly-lactic acid (PLA) [5,6]. Lactic acid (LA) can be separated and purified from various food and agriculture byproducts [7,8]. Then, LA can be converted to PLA through various polymerization processes, including direct polycondensation (DPC), reactive oxygen polymerization (ROP), direct azeotropic dehydration, and enzymatic polymerization. Polycondensation is one of the most common polymerization processes used to manufacture PLA [9,10,11]. Thus, the low-cost accessible preparation of PLA makes it one of the most favorable biopolyesters for packaging and medical applications [12,13,14]. In the packaging field, it can act as a viable substitute for traditional plastics for packaging uses such as trays, bottles, and cups [15]. However, its poor viscoelastic behavior and extremely low melting strength are inhibiting factors for film packaging production. To overcome these disadvantages and increase its ductility and flexibility, PLA is blended with plasticizers. To maintain PLA’s environmentally friendly profile, plasticizers such as epoxidized soybean oil, cardanol, triethyl citrate (TEC), and acetyl tributyl citrate (ATBC) have been successfully used [16,17,18,19,20,21]. Recently, TEC has been used successfully to plasticize PLA, and it has been shown for the first time that it also has self-healing, antioxidant, and antibacterial properties [22]. These innovative properties of TEC make PLA/TEC composite films favorable for active antibacterial flexible food packaging films [23]. 

On the other hand, in the context of bioeconomy and sustainability trends, the Food Technology sector is trying to (i) replace synthetic food preservatives with natural abundant antioxidant/antibacterial compounds such as phytochemicals, natural extracts, and essential oils (EOs) and (ii) incorporate such novel green natural preservatives in packaging to control their release in food. EOs have been shown to be the most favorable candidates among natural preservatives and phytochemicals to be used in antimicrobial active food packaging applications [24,25,26,27,28,29,30,31]. EOs are considered to be safe by the FDA (Food and Drug Administration) and are known as GRAS (General Recognize as Safe) [32]. To overcome their direct loss through the evaporation process, new technologies have been proposed, such as their adsorption onto natural adsorbents such as nanoclays, natural zeolites, and activated carbons, followed by their incorporation in a polymer matrix to develop an active packaging film with the controlled release of EOs in food [33,34,35,36,37,38]. Natural zeolite (NZ) is more favorable for use as a nanocarrier host of EOs than nanoclays due to its higher surface area [36,39,40]. Unlike activated carbons, NZ is more favorable for use as a host for EOs due to its edible properties [41]. Carvacrol (CV) is the main component of thyme and oregano oil, and it is well known for its antioxidant, antibacterial, antitumor, antimutagenic, antigenotoxic, analgesic, antispasmodic, anti-inflammatory, angiogenic, antiparasitic, and antiplatelet activity [42]. In the last few years, various studies have been conducted on the application of CV in active food packaging [43,44,45,46,47]. Recently, CV was loaded on halloysite nanoclays and was then homogeneously distributed as an active coating on the polyethylene surface [47]. The as-prepared coatings present strong antibacterial activity against *Aeromonas hydrophila* and reduce bacterial growth on packaged chicken surfaces [47].

The aim of this work was to employ all the abovementioned advantages of the four natural and/or biodegradable materials, i.e., PLA, TEC, NZ, and CV, to develop a novel active food packaging film that exhibits strong antioxidant/antibacterial, barrier, and mechanical properties. The enhancement of the antioxidant/antibacterial properties of such films extends the shelf life of valuable food and causes no environmental issues.

## 2. Materials and Methods

### 2.1. Materials

PLA with the trade name Ingeo™ Biopolymer 3052D, crystalline melt temperature of 145–160 °C, and glass transition temperature of 55–60 °C was purchased from NatureWorks LLC (Minnetonka, MN, USA). Liquid triethyl citrate (TEC) with an M_w_ of 276.3 g/mol was purchased from Alfa Aesar GmbH & Co KG, (Karlsruhe, Germany). 2,2-Diphenyl-1-picrylhydrazyl (DPPH) was purchased from Sigma-Aldrich (Darmstadt, Germany). Ethanol absolute for analysis and acetate buffer (CH_3_COONa·3H_2_O) were purchased from Merck. Edible natural zeolite was purchased from a local pharmacy market. Fresh minced pork meat from the hind leg of the animal with a size of 5 mm and 12 to 20%wt. fat content was provided by the local meat processing plant “Aifantis Company” within one hour of the slaughter. 

### 2.2. Preparation of CV@NZ Nanohybrids

The CV@NZ nanohybrid was prepared as follows: First, 5 g of as-received NZ were placed in a spherical glass flask and heated for 30 min at 100 °C under 3 bar of vacuum (see Figure 1). Under these conditions, all the adsorbed water was removed from NZ, and the initial light beige color of NZ was turned to dark beige. After the end of the dry cleaning process, the security valve of the pump was closed, and the safety valve of the CV tank was opened to allow the carvacrol to be incorporated dropwise and under stirring into the glass spherical flask (see Figure 1). The obtained CV@NZ nanohybrid was removed and kept for further characterization and use.

### 2.3. CV@NZ Nanohybrid Characterization

#### 2.3.1. XRD and FTIR Characterization of CV@NZ Nanohybrids

The obtained CV@NZ as well as as-received NZ nanohybrids were physiochemically characterized by XRD analysis and FTIR spectrometry. The XRD measurements were made in the range of 0.5–30 2 theta for both CV@NZ and NZ powders using a Brüker XRD D8 Advance diffractometer (Brüker, Analytical Instruments, S.A., Athens, Greece). FTIR analysis measurements were carried out in the range of 400 to 4000 cm^−1^ with an FT/IR-6000 JASCO Fourier-transform spectrometer (JASCO, Interlab, S.A., Athens, Greece). 

#### 2.3.2. Desorption Kinetic Studies of CV@NZ Nanohybrids

Desorption kinetics were performed to determine the %*w*/*w* total amount of CV adsorbed onto NZ, the type of adsorption, and the adsorption enthalpy (ΔH_ads,CV_). The desorption of CV from the obtained CV@NZ nanohybrids was studied using a moisture analyzer AXIS AS-60 (AXIS Sp. z o.o. ul. Kartuska 375b, 80–125 Gdańsk, Poland) as follows: Approximately 150 mg (m_0_) of the CV@NZ nanohybrid was spread in the inner disk of the moisture analyzer, and its weight (m_t_) was recorded as a function of time at 223, 243, 263, and 283 °K. The desorption experiment was repeated three times for each of the above temperatures. Then, by using the recorded m_t_ and t values, the desorption isotherms were constructed by plotting the values of (1 − m_0_/m_t_) as a function of time, and the obtained plots were fitted by using the well-known pseudo-second order adsorption–desorption equation [48]:(1)$$\frac{t}{q_{t}}=\frac{1}{{q_{e}^{2}k}_{2}}+\frac{t}{q_{e}}$$
where k_2_ is the rate constant of the pseudo-second order kinetic model (s^−1^), q_e_ is the desorption capacity at equilibrium (mg desorbate/g desorbent), and q_t_ is the desorption capacities at time t (mg desorbate/g desorbent). 

By fitting the acquired experimental adsorption data into the pseudo-second order kinetic Equation (1), the k_2_, and q_e_ mean values were calculated for all the selected temperatures (223, 243, 263, and 283 °K). Subsequently, by plotting the obtained ln(1/k_2_) values with (1/T), we calculated the desorption energy (Ε^0^_des_) according to the Frenkel equation [49,50,51]:(2)$$\frac{1}{k_{2}}=\frac{1}{A}e^{\frac{E_{des}^{0}}{RT}}$$
and its linear transformed type:(3)$$\ln\left(\frac{1}{k_{2}}\right)=\frac{1}{A}+\frac{E_{des}^{0}}{RT}$$
where k_2_ is the rate constant of the pseudo-second order kinetic model (s^−1^), Ε^0^_des_ is the desorption energy, and A is the Arrhenius constant.

### 2.4. Experimental Design—Preparation of Extruded PLA/TEC/xNZ and PLA/TEC/xCV@NZ Nanocomposite Pellets

The obtained CV@NZ was used as a nanofiller at 5, 10, and 15%wt. content in the PLA/TEC matrix to obtain PLA/TEC/xCV@NZ (x = 5, 10, and 15) nanocomposite pellets. For comparison, pure NZ at 5, 10, and 15%wt. was used as a nanofiller in the PLA/TEC matrix to obtain PLA/TEC/xNZ (x = 5, 10, and 15) nanocomposite pellets. The PLA/TEC matrix used was obtained by blending PLA with 15% *v*/*w* TEC content. The content of TEC chosen was the optimum one according to a recent publication [22].

For the development of all PLA/TEC/xNZ and PLA/TEC/xCV@NZ nanocomposite pellets, a Mini Lab twin-screw extruder (Haake Mini Lab II, Thermo Scientific, ANTISEL, S.A., Athens, Greece) was employed under the following operating conditions: speed of 120 rpm and temperature of 180 °C. The sample code names, the contents of PLA, TEC, pure NZ, and the CV@NZ nanohybrid, and the twin extruder operating conditions (temperature and speed) used for the development of all PLA/TECx composite blends are listed in Table 1 for comparison.

### 2.5. PLA/TEC/xNZ and PLA/TEC/xCV@NZ Film Formation

All the PLA/TEC/xNZ and PLA/TEC/xCV@NZ nanocomposite pellets obtained after the extrusion process as well as pure PLA/TEC pellets were thermomechanically formed into films through a heat pressing process by using a hydraulic press with heated platens (Specac Atlas™ Series Heated Platens, Specac, Orpinghton, UK). Approximately 1.0 g of pellets at a constant pressure of 0.5 MPa and temperature of 180 °C was added to obtain films with an average diameter of 11 cm and a thickness of 0.05–0.11 mm. 

### 2.6. Physicochemical Characterization of PLA/TEC/xNZ and PLA/TEC/xCV@NZ Films

All the obtained PLA/TEC/xNZ and PLA/TEC/xCV@NZ films as well as the PLA/TEC film were characterized by XRD analysis measurements using a Brüker XRD D8 Advance diffractometer (Brüker, Analytical Instruments, S.A., Athens, Greece). Films with 2 cm diameter were oriented on the sample chamber space and were measured at 0.5°–40° 2 theta. FTIR analysis measurements were carried out in the range of 4000–400 cm^−1^ with an FT/IR-6000 JASCO Fourier-transform spectrometer (JASCO, Interlab, S.A., Athens, Greece). For the FTIR measurements, 300 mg of KBr granules were mixed with 30 mg of granulated PLA/TEC/xNZ and PLA/TEC/xCV@NZ films as well as a PLA/TEC film and pressed in a hydraulic press to obtain the samples for FTIR measurements (Pike technologies CRUSHIR digital hydraulic press). Scanning electron microscopy (SEM) images were carried out using a JEOL JSM-6510 LV SEM Microscope (JEOL Ltd., Tokyo, Japan). To enhance the electrical conductivity of samples before SEM analysis, all the samples were coated under vacuum with a gold/palladium (Au/Pd) thin layer (4–8 nm) in a sputtering machine (SC7620, Quorum Technologies, Lewes, UK). The tensile properties were measured according to the ASTM D638 method using a Simantzü AX-G 5kNt instrument (Simantzü. Asteriadis, S.A., Athens, Greece). Three to five dog bone-shaped samples of all PLA/TEC/xNZ and PLA/TEC/xCV@NZ films as well as the PLA/TEC film were tensioned, and the stress–strain values were recorded with the applicable software (TrapeziumX version 1.5.6, Simantzü, Asteriadis, S.A., Athens, Greece). The films’ dynamic mechanical behaviors were examined using a dynamic mechanical analyzer (DMA Q800, TA Instruments, 159 Lukens Drive New Castle, DE, USA) in film tension mode. To evaluate the storage modulus (E′), a temperature range of −20 °C to 60 °C at a rate of 5 K/min, along with a frequency of 1 Hz, was applied. Three to five rectangular-shaped films were tensioned for all PLA/TEC/xNZ and PLA/TEC/xCV@NZ films as well as the PLA/TEC film.

### 2.7. Characterization of the Active Packaging Properties of Obtained PLA/TEC/xNZ and PLA/TEC/xCV@NZ Films

#### 2.7.1. Water Barrier Properties

Water barrier properties were determined at 38 °C and 95% RH according to the ASTM E96/E 96M-05 method for three to five samples of all obtained PLA/TEC/xNZ and PLA/TEC/xCV@NZ films as well as the PLA/TEC film. A handmade apparatus was used according to the methodology described previously [52,53]. The obtained Water Vapor Transmission Rate (WVTR) values were transformed to water vapor diffusion coefficient values (D_wv_) by using Fick’s low theory and the following equation:(4)$$D_{WV}=WVTR\cdot \frac{Dx}{DC}$$
where WVTR [g/(cm^2^·s)] is the water vapor transmission rate, Dx (cm) is the film thickness, and DC (g/cm^3^) is the humidity concentration gradient on the two opposite sides of the film.

#### 2.7.2. Oxygen Barrier Properties

The oxygen transmission rate (OTR) values of all obtained PLA/TEC/xNZ and PLA/TEC/xCV@NZ films as well as the PLA/TEC film were measured by using an oxygen permeation analyzer (O.P.A., 8001, Systech Illinois Instruments Co., Johnsburg, IL, USA). Three films of each sample with an average diameter of 11 cm and a thickness of 0.05–0.11 mm were placed in the oxygen analyzer chamber and measured at 23 °C and 0% RH according to the ASTM D 3985 method. After the end of each measurement, each sample was cut into three pieces, and the thickness was measured at four points for each piece. Thus, the thickness of each film was determined by twelve different points, and the average value of thickness was calculated. This mean value was used to transform the obtained OTR values to oxygen diffusion coefficient (P_O2_) values using the methodology previously described [52]. 

#### 2.7.3. CV Release Kinetics—Calculation of Released CV %wt., Total CV Content, and CV Released Diffusion Coefficient (D_CV_)

The kinetics of CV release from all the obtained PLA/TEC/xCV@NZ active films were carried out using a moisture analyzer AXIS AS-60 (AXIS Sp. z o.o. ul. Kartuska 375b, 80-125 Gdańsk, Poland) and the methodology described previously [40]. For the release kinetic experiments, 600 to 900 mg of each film was used in triplicate. For each film, at least three samples were measured. Each film was placed inside the moisture analyzer, and its mass was monitored by heating at 70 °C for 1 h. Using the obtained film mass values (m_t_) as a function of time (t), for each film, the mean values of CV release rate (RR_CV_) as well as the mean values of %wt. CV released content (%RC_CV_) were determined. 

Next, by employing the pseudo-second order sorption mechanism model described above, the constant k_2_ of the CV desorption rate and the maximum CV desorbed amount q_e_ were calculated according to Equation (1).

#### 2.7.4. In Vitro Antioxidant Activity Determination of the Obtained PLA/TEC/xNZ and PLA/TEC/xCV@NZ Films

For the preparation of [DPPH^•^] free radical standard solutions, 0.0212 g of [DPPH^•^] free radical was dissolved in 250 mL of methanol to obtain a 2.16 mM (mmol/L) methanolic solution. Next, the flask was vortexed under dark conditions, and its pH (Milwaukee MW102-FOOD PRO+ 2-in-1 pH and Temperature Meter) was measured to ensure its neutrality (7.02 ± 0.01). Finally, the solution was placed in a refrigerator at 4 ± 1 °C under dark conditions for stabilization.

For the preparation of a [DPPH^•^] free radical calibration curve, 2.16 mM (mmol/L) methanolic solution of [DPPH^•^] free radical was diluted by adding appropriate volumes of methanol to obtain concentrations of 10, 20, 30, 40, and 50 mg/L, and their absorbance was measured with a SHIMADZU UV-1280 UV/VIS Spectrometer at 517 nm. The calibration curve of absorbance (y) versus the concentration (x) of [DPPH^•^] free radical was expressed by the following equation:y = 0.0388x + 0.015; R^2^ = 0.9994(5)

For the determination of the concentration required to obtain a 50% antioxidant effect (EC_50_) from all the obtained PLA/TEC/xNZ and PLA/TEC/xCV@NZ films as well as the PLA/TEC film, 10, 20, 30, 40, and 50 mg of granule film were placed in dark vials and three replicates were performed for each sample. Thereafter, 3 mL of [DPPH^•^] free radical methanolic solution and 2 mL of acetate buffer 100 mM (pH = 7.10) were added to each vial, and the absorbance of the reaction mixture was measured at 517 nm after 24 h. For a blank sample, we used a vial containing 3 mL of [DPPH^•^] free radical methanolic solution and 2 mL of acetate buffer without the addition of any granule film. The % inhibition of [DPPH^•^] was calculated using the following equation:(6)$$\%\;scavenged\;{DPPH}^{•}\;at\;steady\;state=\frac{A_{0}^{517}-A_{sample}^{517}}{A_{0}^{517}}\times 100$$

#### 2.7.5. Antibacterial Activity of the Obtained PLA/TEC/xCV@NZ Films

The antimicrobial activity of all obtained PLA/TEC/xCV@NZ nanocomposite films as well as the PLA/TEC composite film was tested against one Gram-positive *Staphylococcus aureus* (NCTC 6571) food pathogen and one Gram-negative *Salmonella enterica* subspecies enterica serovar *Typhimurium* (NCTC 12023) food pathogen [22]. The bacteria were supplied by Supelco^®^ Analytical Products, a subsidiary of Merck (Darmstadt, Germany) as microbiological certified reference materials in the form of easy-tab™ pellets (*S. aureus*) by LGC Standards Proficiency Testing (Chamberhall Green Bury, Lancashire, UK) and in the form of disc-shaped Vitroids™ (*S. Typhimurium*).

The experimental procedure followed was based on the agar diffusion method described recently [55]. Briefly, a concentration of approx. 1.5 × 10^8^ CFU/mL (i.e., 0.5 McFarland standard) per bacterial strain was achieved in 3 mL of 0.85% peptone salt solution or maximum recovery diluent (MRD; Merck). The suspended bacteria in MRD were inoculated on Müeller-Hinton (MH) agar (Oxoid, Basingstoke, UK) plates with the help of sterile cotton applicators (Jiangsu Kangjin Medical Instrument Co., Taizhou, China). Samples with a diameter of 6 mm of all PLA/TEC/xCV@NZ nanocomposite films as well as the PLA/TEC composite film were placed onto the surface of the MH agar, and the incubation took place at 37 °C for 18–24 h. The diameters of inhibition zones within and around the contact area of the films were measured by using a Vernier caliper, with 0.1 mm accuracy. The experimental procedure was repeated twice, while films were measured in triplicate in each repetition.

### 2.8. Packaging Test of Fresh Minced Pork Wrapped in PLA/TEC and PLA/TEC/10CV@NZ Films

#### 2.8.1. Packaging Preservation Test of Minced Pork Meat

Minced pork from the hind leg, in portions of approximately 70–80 g each, were aseptically wrapped between two films of PLA/TEC and PLA/TEC10CV@NZ samples that were 11 cm in diameter and placed inside the Aifantis company’s commercial wrapping paper without the inner film (coated with plasticized PVC). For the control sample, 80–90 g of minced pork was aseptically wrapped in the commercial opaque packaging paper (without removing the inner coated PVC film) from the Aifantis company. For all tested packaging systems, samples for the 2nd, 4th, 6th, 8th, and 10th day of preservation were prepared and stored under dark refrigerator conditions at 4 ± 1 °C (LG GC-151SA, Weybridge, UK). Every two days until the 10th day of storage, lipid oxidation, heme iron content, total viable count, and sensory analysis scores were determined for every packaging treatment.

#### 2.8.2. Lipid Oxidation of Minced Pork Meat with Thiobarbituric Acid Reactive Substances

The rates of lipid oxidation of all samples of minced pork during the 10 days of storage were calculated with the thiobarbituric acid reactive substances (TBARS) method according to Tarladgis et al. [56]. In brief, 2 g of meat sample was placed in a vial, along with 5 mL of a 10% (*w*/*v*) trichloroacetic acid (TCA) solution, and vortexed for 5 min. Then, 5 mL of 0.02 M 2-thiobarbituric acid aqueous solution was added to it and vortexed a second time for 5 min. After that, the obtained mixture was left in the dark for 18–24 h for color development. Finally, the obtained colored mixture was centrifuged and the absorbance (D) in the resultant supernatant was measured at λ = 538 nm. For the blank sample, 5 mL of the 10% (*w*/*v*) TCA solution and 5 mL of a 0.02 M aqueous solution were mixed, and the absorbance was also measured at λ = 538 nm. 

TBARS was expressed as mg of malondialdehyde (MDA)/kg of the sample according to the following equation:TBARS (mg/kg) = 3.6 × D(7)

#### 2.8.3. Heme Iron Content

The values of heme iron content of all samples of minced pork during the 10 days of storage were determined according to the method reported by Clark et al. [57]. Briefly, 4 g of minced pork meat was dispersed in 18 mL of acidified acetone and homogenized with a mixer (Vicko S.A, Athens, Greece). The solution was then filtered after standing at 25 °C under dark conditions for 1 h by using 45 nm Whatman UK filters. Finally, in the filtered solution, the absorbance was measured at 640 nm by using a UV-vis spectrophotometer (SHIMADZU UV-1280, SHIMADZU, Kyoto, Japan), and the heme iron content in minced pork was calculated using the Equation (8):Heme iron (μg/g) = A_640_ × 680 × 0.0882(8)
where A_640_ is the absorbance measured at 640 nm and 680 and 0.0882 are constant values in the equation.

#### 2.8.4. Total Viable Count (TVC) of Minced Pork Meat

For the estimation of total viable count (TVC) for all samples of minced pork during the 10 days of storage, a recently described methodology was followed [55]. Briefly, 10 g of pork fillets was aseptically removed from each packaging system and homogenized with 90 mL of sterile buffered peptone water (ΒPW, NCM0015A, Heywood, UK) at room temperature. For the microbial enumeration, 0.1 mL of serial dilutions (1:10 diluents, buffered peptone water) of homogenized minced pork products were spread on the surface of plate count agar (PCA, NCM0010A, Heywood, UK). 

#### 2.8.5. Sensory Analysis of Minced Pork Meat

Sensory properties like color, odor, and texture were evaluated during the 10 days of storage. At each sampling day (2nd, 4th, 6th, 8th, and 10th day of storage), for each packaging treatment, all the above properties were ranked from 0 (lowest degree of each characteristic in the tested samples) to 5 (highest degree of each characteristic in the tested samples) by seven experienced panelist/members of the Department of Food Science and Technology experienced in meat sensory evaluation by using the conventional descriptive analysis [58,59,60]. A descriptive analysis panel determined the important product characteristics and then evaluated the degree of each characteristic in the tested samples.

### 2.9. Statistical Analysis

All the data acquired from structural and mechanical properties measurements, along with antioxidant activity, thiobarbituric acid reactive substances, heme iron content, total viable count, and sensory analysis scores, were subjected to statistical analysis to indicate any statistical differences. Mood’s median test method, a non-parametric statistical procedure, was chosen. All measurements were conducted on a minimum of three to five separate samples, and the least significant difference was that of *p* < 0.05. Among the three packaging treatments (commercial paper (control), PLA/TEC, and PLA/TEC/10CV@NZ), a correlation between TBARS and heme iron content values was estimated using Pearson’s bivariate correlation (−1 to +1) at the confidence level *p* < 0.05 with respect to storage time (see Appendix A). Statistical analysis was conducted using SPSS software (v. 28.0, IBM, Armonk, NY, USA).

## 3. Results

### 3.1. Physicochemical Characterization of the Obtained CV@NZ Nanohybrids

In Figure 2a, the XRD plots of pure NZ and the modified CV@NZ nanohybrid are shown for comparison. 

The observed reflections in the patterns of both pure NZ and the TO @NZ nanohybrid are assigned to the Heulandite Ca(Si_7_Al_2_)O_16 ×_ 6H_2_O monoclinic crystal phase (PDF-41-1357). Thus, in accordance with a previous report, the adsorption of CV onto NZ did not affect the crystallinity of pure NZ [61].

In the FTIR spectrum of CV (see plot line (1) in Figure 2b) the absorption bands in the range of 1521–1600 cm^−1^ are assigned to C=C bond stretching vibrations of the aromatic ring [62,63]. Furthermore, the bands at 1361 and 1382 cm^−1^ are assigned to the symmetric and asymmetric isopropyl methyl group vibrations, respectively [62,63]. The broadband at 3398 cm^−1^ can be attributed to the stretching vibration of the O–H functional group of the CV molecule. The bands appearing at 812 and 866 cm^−1^ are the characteristic bands of C–H out-of-plane wagging vibrations [62,63]. The absorption bands at 2875–3021 cm^−1^ are assigned to the stretching vibrations of aliphatic C–H groups, and the bands in the range of 1066–1117 cm^−1^ are attributed to the ortho-substituted phenyl group [62,63].

In the FTIR plot of pure NZ (see plot line (2) in Figure 2b), the bands at 3619 and 3436 cm^−1^ are assigned to the OH group stretching mode, the band at 1650 cm^−1^ is assigned to the OH group bending mode, the band at 1090 cm^−1^ is assigned to the Si-O stretching vibration, and the band at 468 cm^−1^ is assigned to the -SiO- bending mode [61,64,65]. As shown in Figure 2b, the obtained FTIR plot of CV@NZ is a combination of pure CV and pure NZ reflections (see dot line pink box of plot line (3) in Figure 2b). This suggests the effective adsorption of CV molecules onto NZ pores. Moreover, in the obtained FTIR plot of the CV@NZ nanohybrid, there are indications of chemisorption of CV onto NZ. These are (i) the band’s widening at 3436 cm^−1^ of the NZ hydroxyl group, along with the band at 3398 cm^−1^ of hydroxyl groups of CV and the red shift in the band at 3619 cm^−1^ of the hydroxyl group of NZ. 

In Figure 3, the CV desorption isotherm kinetic plots (scatter plots) and the simulation plot lines (red line plots) are compared. 

The plots in Figure 3 were simulated with a pseudo-second order kinetic model according to Equation (1) and the calculated mean values of k_2_, pseudo-second order constant k_2_, and the desorption capacity at equilibrium q_e_ are all shown in Table 2 for comparison. Table 2 also includes the calculated mean values of %wt. CV desorbed at 223 °K, 243 °K, 263 °K, and 283 °K.

As shown in Table 2 the total adsorbed amount of CV onto the CV@NZ nanohybrid is equal to 61.7 ± 0.23%wt. This amount is much higher than the value of 35.5%wt. recently reported as the adsorbed amount of thymol onto NZ [40]. This increment must be attributed to the vacuum adsorption method employed here. The clean-vacuum process of pure NZ led to the desorption of all adsorbed water molecules of NZ and increased the available NZ adsorption active sites for CV.

Based on the average values of k_2_ and q_e_ reported in Table 2, it follows that with increasing temperature, the value of k_2_ increases, while the value of q_e_ decreases. 

By plotting the obtained ln(1/k_2_) values with (1/T) in Figure 4, we calculated the desorption energy (Ε^0^_des_) according to the Frenkel Equation (2) and the linear equation in Figure 4.

Τhe CV desorption energy (E^0^_des_) was found to be 90.32 KJ/mol, which is equal to 21.58 Kcal/mol. According to the Arrhenius theory, E^0^_des_ values higher than 20 Kcal/mol correspond to chemisorption [49,50,51]. In our study, the calculated value of 21.58 Kcal/mol was close to the threshold of 20 Kcal/mol, which corresponds to mixed chemisorbed and physiosorbed adsorption of CV onto NZ. This mixed chemisorbed and physiosorbed adsorption of CV onto NZ is in accordance with the pseudo-second order of desorption kinetics and the FTIR results discussed above, suggesting the chemisorption of CV molecules via an interaction of CV’s OH groups with NZ’s OH groups. 

### 3.2. Physicochemical Characterization of PLA/TEC/xNZ and PLA/TEC/xCV@NZ Films

Figure 5a shows the XRD plots of all obtained PLA/TEC/xNZ and PLA/TEC/xCV@NZ films as well as the PLA/TEC film in the range of 0.5° to 40° 2theta for comparison.

As shown in Figure 5a, the addition of both pure NZ and the CV@NZ nanohybrid in the PLA/TEC matrix results in the appearance of small peaks in the amorphous crystal phase plot of PLA/TEC. These reflections correspond to the reflections of NZ. At first glance, such NZ reflections are more intensive in the case of all PLA/TEC/xNZ films than in the case of PLA/TEC/xCV@NZ films, and based on recent reports, it suggests that the CV@NZ nanohybrid in the PLA/TEC matrix has higher dispersion than pure NZ [40,61].

In plot line (1) of Figure 5b where the FTIR spectrum of PLA/TEC is shown, the following characteristic absorption bands are observed: the O–H stretching vibration at 3450 cm^−1^, the asymmetrical and symmetrical –C–H stretching vibrations of the methyl group in the side chains at 2950, 3000, and 3050 cm^−1^, the C = O stretching vibration carboxyl group from the repeated ester unit at 1760 cm^−1^, the–C–O– stretching vibrations in the range of 1110–1330 cm^−1^, and the corresponding bending vibrations of the –CH_3_ group at 1451 cm^−1^ [22].

When both pure NZ and the CV@NZ nanohybrid are added in the PLA/TEC matrix, the characteristic bands of NZ are added in the FTIR plots of PLA/TEC/xNZ and PLA/TEC/xCV@NZ films. More specifically, they are observed at the band at 468 cm^−1^, which corresponds to the -SiO- bending mode, and the band at 1650 cm^−1^, which corresponds to the OH group bending mode [40,61]. At the same time, it was observed that all the characteristic reflections of PLA/TEC FTIR are increased. This increment is more observable for the reflection of the O–H stretching vibration of PLA/TEC at 3450 cm^−1^. Overall, the FTIR plots show that both NZ and CV@NZ are effectively mixed with the PLA/TEC matrix, resulting in interactions with both the hydroxyl and aliphatic chains of the PLA/TEC matrix. This result follows previous reports and suggests NZ and modified CV@NZ as an excellent reinforcement material for both polymer and biopolymer-based matrices [40,61].

Scanning electron microscopy (SEM) was used to investigate the material’s behavior based on the effect of the nanohybrid level on films and the surface features. Thus, the morphological characteristics of the surface and cross-sections of film samples were evaluated with SEM and are presented in Figure 6. 

The surface and cross-section images of the PLA/TEC polymer matrix and hybrid nanocomposite films of PLA/TEC/NZ and PLA/TEC/CV@NZ with different concentrations (5, 10, and 15) of the CV@NZ hybrid nanostructure and pure NZ (white dots can be clearly identified on the surface of the PLA/TEC matrix) are shown in Figure 6. All obtained films were clear and uniform in thickness. The surface/cross-section of the polymer matrix PLA/TEC film (Figure 6a,b) revealed smooth and clear surface morphology with a continuous phase without heterogeneities. 

Surface and relative cross-section images of PLA/TEC/xNZ and PLA/xCV@NZ with different ratios of NZ and CV@NZ are presented in Figure 6c–n. It is evident that increasing the contents (after the incorporation of NZ and CV@NZ) in the nanocomposites increases the degree of dispersion accordingly. Nevertheless, the results of SEM studies of the final nanocomposite films confirmed that the nanohybrids were homogeneously dispersed, indicating their enhanced compatibility with the polymer matrix, which was the major parameter that improved the behavior of the films. 

It should be mentioned that based on the SEM studies (surface and cross-section), a significant difference was observed when the CV@NZ hybrid nanostructure is incorporated into the amorphous parts of the PLA/TEC polymer matrix as better interfacial adhesion and homogenous dispersion are evident compared to the corresponding nanocomposite film with pure NZ.

### 3.3. Mechanical and Thermomechanical Properties of PLA/TEC/xNZ and PLA/TEC/xCV@NZ Films

The stress–strain curves of all PLA/TEC/xNZ and PLA/TEC/xCV@NZ films as well as the pure PLA/TEC film are plotted in Figure 7. From these stress–strain curves, the Elastic Modulus (E) values, the ultimate strength (σ_uts_) values, and the % elongation at break (%ε) values were calculated and are listed in Table 3 for comparison. Independent sample median tests and pairwise comparisons of the different treatments according to the mean values of E, σ_uts_, and %ε are shown in Appendix A, respectively.

For PLA/TEC/xNZ films, as the NZ %wt. content increases, the ultimate strength values increase and the %ε decreases compared to the PLA/TEC film, as shown by the values reported in Table 3. In the case of PLA/TEC/xCV@NZ films, the addition of CV@NZ results in an increase of obtained %ε values compared to pure PLA/TEC. For 5%wt. CV@NZ nanohybrid addition, the obtained PLA/TEC/5CV@NZ film has higher ultimate strength values than both pure PLA/TEC and PLA/TEC/5NZ films. For 10%wt. CV@NZ nanohybrid addition, the obtained PLA/TEC/10CV@NZ film has an ultimate strength value approximately equal to the pure PLA/TEC film and lower than the PLA/TEC/10NZ film. For 15%wt. CV@NZ nanohybrid addition, the obtained PLA/TEC/15CV@NZ film has much lower ultimate strength values than the PLA/TEC film. Overall, it can be concluded that pure NZ acts as a reinforcement agent and reinforces the PLA/TEC matrix when it is added in low, medium, and higher contents. On the other hand, the role of the CV@NZ nanohybrid appears to be more of a plasticizer than a reinforcement agent. Considering that the aim of adding the CV@NZ nanohybrid to the PLA/TEC matrix is to load as much as CV %wt. as we can without destroying its tensile properties, we concluded that 10%wt. addition CV@NZ is optimal. 

The dynamic mechanical analysis (DMA) measurements of the storage modulus for all tested films of the PLA/TEC polymer matrix and hybrid nanocomposite films of PLA/TEC/NZ and PLA/TEC/CV@NZ with different concentrations of NZ and hybrid nanostructure CV@NZ are summarized in Figure 8. 

As shown in Figure 8, the integration of NZ in the PLA/TEC matrix decreased the storage modulus at all concentrations used (5, 10, and 15%). The incorporation of the hybrid nanocomposite CV@NZ, and in particular, the concentrations of 10% and 15%, improved the thermomechanical behavior of the samples, perhaps due to better dispersion of the hybrid nanostructure in the PLA/TEC matrix.

### 3.4. Water/Oxygen Barrier Properties of PLA/TEC/xNZ and PLA/TEC/xCV@NZ Films

In Appendix A, the obtained water vapor transmission rate (WVTR) and oxygen transmission rate (OTR) mean values of all tested PLA/TEC/xNZ and PLA/TEC/xCV@NZ films and the pure PLA/TEC film are listed for comparison. From these values, the water vapor diffusion coefficient (D_wv_) values and the oxygen permeability P_eO2_ values were calculated and are listed in Appendix A for comparison. In Appendix A, the independent sample median test and pairwise comparisons of all the obtained films according to the mean values of Dwv and P_eO2_ are shown. In Figure 9, the obtained D_wv_ and P_eO2_ mean values of all tested PLA/TEC/xNZ and PLA/TEC/xCV@NZ films as well as for the pure PLA/TEC film are plotted for comparison.

As shown in Figure 9, the addition of both pure NZ and the CV@NZ nanohybrid led to an increase in both water/oxygen barrier properties of all obtained PLA/TEC/xNZ and PLA/TEC/xCV@NZ films compared to the water/oxygen barrier of the pure PLA/TEC film. This result agrees with the high dispersion of both pure NZ and the CV@NZ hybrid in obtained PLA/TEC/xNZ and PLA/TEC/xCV@NZ films, correspondingly derived from the above SEM images. The high dispersion of a nanofiller is known to be beneficial for the water/oxygen barrier properties of polymer nanocomposites because it reduces the free paths available for water/oxygen to permeate through the polymer matrix [66,67]. In the case of PLA/TEC/xNZ films, the water/oxygen barrier increases as the NZ %wt. content increases. So, the lowest D_wv_ and Pe_O2_ values are obtained for the PLA/TEC/15NZ film. In the case of PLA/TEC/xCV@NZ films, the water/oxygen barrier decreases as the CV@NZ %wt. content increases. Thus, the lowest D_wv_ and Pe_O2_ values are obtained for the PLA/TEC/5CV@NZ film.

### 3.5. CV Release Kinetics of PLA/TEC/xCV@NZ Films

In Appendix A, the obtained (1 − m_t_/m_o_) values are plotted as a function of time for each film for the simulation with pseudo-second order based on Equation (1). From the simulation of experimental values obtained with Equation (1), the mean values of the CV desorption equilibrium constant (q_e_) and the desorption rate constant mean values (k_2_) were calculated and are listed in Table 4 for each film for comparison. Table 4 also shows the calculated mean values of the CV release rate (RR_CV_) as well as the mean values of %wt. values of CV released content (%RC_CV_) for each film for comparison.

The data in Table 4 show that both the k_2_ mean values and RR_cv_ mean values decrease as the CV@NZ content increases. This means that as the CV@NZ content increases, the CV release rate decreases. On the other hand, both the CV desorption equilibrium constant (q_e_) mean values and the %RC_CV_ mean values increase as the CV@NZ content increases. This means that as the CV@NZ content increases, the CV release amount also increases. In other words, by increasing the CV@NZ content, the total amount of CV release content increases, but the CV release rate decreases. This means that medium CV@NZ content (10CV@NZ) is optimal for obtaining a high CV release content and CV control release rate. 

### 3.6. Antioxidant Activity of PLA/TEC/xCV@NZ Films

Table 5 shows the calculated EC50 mean values for all obtained PLA/TEC/xCV@NZ active films. In Appendix A, the plots and the linear equations used to calculate the mean EC50 values for each PLA/TEC/xCV@NZ film are presented. In Appendix A, the independent sample median test and pairwise comparisons of all the obtained films according to the mean values of EC50 are listed.

As shown in Table 5, all PLA/TEC/xCV@NZ active films exhibited a low EC50 value in the range of 2.85 to 4.65 mg/L. These EC50 values are much lower than 212.6 mg/L, which is the EC50 value of pure PLA/TEC that was recently reported [22]. In addition, it was observed that all PLA/TEC/xCV@NZ films exhibited low and equal EC50 values that were not significantly different (see Appendix A). This result agrees with the CV release kinetics discussed above. It was shown that by increasing the CV@NZ content, the total amount of CV release content increases, but the CV release rate decreases. So, it seems that all PLA/TEC/xCV@NZ films release equal amounts of CV and reach equal EC50 values during the time of incubation for the antioxidant activity experiment. 

### 3.7. Antibacterial Activity of PLA/TEC/xCV@NZ Films

The results of the antibacterial activity of all PLA/TEC/xCV@NZ films against *S. aureus* and *S. Typhimurium* are presented in Table 6.

The values in Table 6 show that all PLA/TEC/xCV@NZ films exhibited significant antibacterial activity against Gram-positive (*Staphylococcus aureus*) and Gram-negative (*Salmonella enterica* ssp. *enterica* serovar Typhimurium) pathogenic bacteria. All the tested films had strong antibacterial activity against both pathogens, i.e., *S. aureus* and *S. Typhimurium*, in the contact area but also created a significant inhibition zone. As recently shown, pure PLA/TEC films exhibited antibacterial activity against the same pathogens only in the contact area [22]. Thus, it can be concluded that the incorporation of the CV@NZ nanohybrid to obtain PLA/TEC/xCV@NZ films not only has a positive effect on the antibacterial activity but also, due to the inhibition zones generated from all the films, it is stronger. As shown in Table 6, no significant differences can be observed for the antibacterial activity of films loaded with different CV@NZ nanohybrid contents. This result is in accordance with the antioxidant activity results of PLA/TEC/xCV@NZ films.

### 3.8. Preservation of Fresh Minced Pork Meat Wrapped in PLA/TEC and PLA/TEC/10CV@NZ Films

#### 3.8.1. Microbiological Evaluation of Minced Pork Wrapped in PLA/TEC and PLA/TEC/10CV@NZ Films

The obtained mean TVC values according to the kind of film used and the storage time are listed in Appendix A. The independent sample median test and pairwise comparisons of all obtained films according to the mean values of TVC are shown in Appendix A. For better comparison, the TVC values are plotted in the column bar diagram in Figure 10a according to the film used and storage time. 

The total viable count (TVC) of bacteria is an important microbiology indicator for the quality and safety of meat [68,69]. The TVC limit of acceptance for fresh pork meat is 7 log CFU/g according to International Commission on Microbiological Specifications for Foods (ICMFS) [56]. When pork meat TVC exceeds this standard value of acceptance, it could greatly endanger consumers’ health [69]. As shown in Figure 10a and Appendix A, minced pork wrapped in the commercial packaging of the Aifantis company and the PLA/TEC film almost exceeded the limit of acceptance after the fourth and sixth day of storage, respectively. The minced pork wrapped in the PLA/TEC/10CV@NZ film almost exceeded that value after the eighth day of storage. According to the obtained values, for the total viable count, both PLA/TEC and PLA/TEC/10CV@NZ films succeeded in extending the microbiological shelf life of fresh minced pork for two and four days, compared to the commercial packaging paper (control), respectively. The two-day shelf life extension of fresh minced pork using the PLA/TEC film is in agreement with a previous report that showed that the incorporation of TEC into the PLA matrix not only extends the shelf life of pork meat for two days but also improves the antioxidant and antibacterial activity of the obtained PLA/TEC film [22]. Extending the shelf life of fresh minced pork for four days using the PLA/TEC/10CV@NZ film is reported for the first time, making it very promising for active packaging. It should be stressed that PLA/TEC/10CV@NZ recorded the lowest (significantly different) values throughout the storage time (see Appendix A). This result is supported by the characterization results of the PLA/TEC/10CV@NZ film where a high dispersion of the CV@NZ nanohybrid in the PLA/TEC matrix was shown, improving the antibacterial activity of the film and the obtained CV’s control release rates.

#### 3.8.2. Lipid Oxidation of Minced Pork Wrapped in PLA/TEC and PLA/TEC/10CV@NZ Films

The 2-thiobarbituric acid reactive substances (TBARS) test is an important quality index for pork meat preservation [70,71]. The TBARS value reflects the content of malonaldehyde, one of the degradation products of lipid hydro peroxides and peroxides formed during the oxidation of polyunsaturated fatty acids, and thus, the TBARS value is widely used as an indicator of the degree of lipid oxidation during the deterioration of pork meat [72]. The obtained mean TBARS values according to the kind of film used and the storage time are listed in Appendix A. The independent sample median test and pairwise comparisons of all the obtained films according to the mean TBARS values are listed in Appendix A. For better comparison, the TBARS values are plotted in the column bar diagram in Figure 10b according to the type of film used and storage time. As shown by the TBARS plots in Figure 10b and Appendix A, all packaging systems had statistically significant differences throughout the storage time, except for the 8th day, at which time the control and PLA/TEC samples had similar values. Minced pork wrapped in both PLA/TEC and PLA/TEC/10CV@NZ films exhibited lower TBARS increment rates than minced pork wrapped in the commercial film (control sample). Minced pork wrapped in the PLA/TEC/10CV@NZ film recorded the lowest TBARS increment rate. Thus, on the 8th day, when minced pork wrapped in the PLA/TEC/10CV@NZ film almost exceeded the limit of acceptance for TVC, 7 logCFU/g, minced pork meat had a TBARS value that was 17% lower than that for minced pork meat wrapped in commercial film. In other words, the PLA/TEC/10CV@NZ film succeeded in preventing minced pork from lipid oxidation deterioration. This result is in accordance with the improved water/oxygen barrier properties and enhanced antioxidant activity of this film, as discussed above.

#### 3.8.3. Heme Iron Content of Minced Pork Wrapped in PLA/TEC and PLA/TEC/10CV@NZ Films

Heme iron content is an important nutritional index for pork and other meats [58,73]. The obtained mean heme iron content values according to the kind of film used and the storage time are listed in Appendix A. The independent sample median test and pairwise comparisons of all the obtained films according to the mean values of heme iron are listed in Appendix A. For better comparison, heme iron content values are plotted in the column bar diagram in Figure 10c according to the type of film used and storage time. As shown in Figure 10c, in all the samples, heme iron content decreases as the storage time increases. According to Appendix A, the control and PLA/TEC samples seemed to have the same trend in the first two days of storage, while PLA/TEC/10CV@NZ was significantly different. On the 4th, 6th, and 8th days of storage, all packaging treatments recorded statistically significant differences. On the 10th day of storage, only minced pork wrapped in PLA/TEC/10CV@NZ films recorded lower and significantly different values for heme iron content. Both PLA/TEC and PLA/TEC/10CV@NZ films succeeded in keeping wrapped minced pork with higher heme iron contents during the ten days of storage compared to minced pork wrapped in commercial paper (control sample). The highest heme iron content values during the ten days of storage were obtained for minced pork wrapped in the PLA/TEC/10CV@NZ film. This result is in accordance with the low TBARS values obtained for the minced pork wrapped in the same packaging system. In other words, PLA/TEC/10CV@NZ prevents minced pork from lipid oxidation deterioration and thus maintains a higher heme iron content in minced pork. According to previous reports, a linear correlation between the increasing TBARS and the decreasing heme iron content values has been recorded [38,40,74]. In this study, a correlation between TBARS and heme iron content values was estimated using Pearson’s bivariate correlation (−1 to +1) at the confidence level *p* < 0.05 with respect to storage time. The results of the statistical analysis showed that there is a statistically significant and positive correlation between the two methods throughout the storage period (see Appendix A).

#### 3.8.4. Sensory Analysis of Minced Pork Wrapped in PLA/TEC and PLA/TEC/10CV@NZ Films

Sensory analysis results of odor, color, and texture of minced pork wrapped in PLA/TEC and PLA/TEC/10CV@NZ films as well as with a commercial film are presented in Table 7 for comparison. The independent sample median test and pairwise comparisons of all the obtained films according to the mean values of odor, color, and texture are listed in Appendix A.

According to statistical analysis results, odor, color, and texture are significantly different for all the tested samples after the 4th day of storage. For the minced pork sample wrapped in the PLA/TEC/10CV@NZ film, odor, color, and texture are significantly different from the other two samples after the 2nd day of storage. Overall, it can be concluded that both pure PLA/TEC films and PLA/TEC/10CV@NZ films succeed in preserving wrapped minced pork in much better and acceptable sensory conditions than commercial packaging paper. Minced pork wrapped in the PLA/TEC/10CV@NZ film exhibited the highest sensory analysis values during the 10 days of storage. It is noteworthy that minced pork packaged with the PLA/TEC/10CV@NZ film has odor, color, and texture values that are higher than three on the 10th day of storage. 

## 4. Discussion

In this study, a novel CV@NZ hybrid nanostructure and PLA/TEC/xNZ, PLA/TEC/xCV@NZ (x = 5, 10, and 15%wt.) nanocomposite active packaging films were successfully developed. The XRD analysis of the CV@NZ nanohybrid showed that the adsorption of CV onto NZ does not affect the crystallinity of pure NZ, while the FTIR analysis suggests the effective adsorption of CV molecules onto NZ pores. Moreover, in the obtained FTIR plot of the CV@NZ nanohybrid, there are indications of chemisorption of CV on NZ. The desorption kinetic experiments of CV from the CV@NZ nanohybrid revealed that the total adsorbed amount of CV onto the CV@NZ nanohybrid is equal to 61.7 ± 0.23%wt. This amount is much higher than the value of 35.5%wt. recently reported as the adsorbed amount of thymol onto NZ [40]. This increment is attributed to the novel vacuum adsorption method employed herein. The results presented herein for the vacuum-assisted adsorption of CV onto NZ are similar to those recently presented for the vacuum-assisted adsorption of cinammaldehyde and peppermint essential oil onto halloysite nanotubes [75,76]. In addition, desorption kinetic experiments showed that the desorption energy of CV is 21.58 Kcal/mol, which suggests mixed chemisorbed and physiosorbed adsorption of CV onto NZ following the FTIR results. It must be stressed that such desorption kinetic experiments are reported for the first time as a novel method to determine the %wt. loading content as well as the desorption energy of CV onto such a CV@NZ nanohybrid. This CV desorption kinetic method is provided as a novel and low-cost method to replace high-cost methods usually used, such as thermogravimetric analysis (TG) and differential scanning calorimetry (DSC) measurements [38,40]. 

For PLA/TEC/xNZ and PLA/TEC/xCV@NZ (x = 5, 10, 15%wt.) nanocomposite active packaging films, both XRD and FTIR suggest higher dispersion of the CV@NZ nanohybrid in the PLA/TEC matrix than pure NZ. SEM studies confirmed that both pure NZ and the CV@NZ nanohybrid were homogeneously dispersed, which indicates their enhanced compatibility with the PLA/TEC matrix. The tensile measurements showed that pure NZ serves to reinforce the PLA/TEC matrix regardless of whether it is added in low, medium, or high amounts. Rhim et al. [77] also added Ag-zeolite in a chitosan biopolymer matrix and concluded that by using Ag-zeolite, the tensile strength of the film can be increased by 7–16% [77]. On the other hand, the role of the CV@NZ nanohybrid seems to be more of a plasticizer agent than a reinforcement agent due to the presence of CV adsorbed onto NZ, which acts as plasticizer [78]. Considering that the aim of this study was to add the CV@NZ nanohybrid to PLA/TEC matrix to gain the highest CV content (%wt.) possible without worsening its tensile properties, we concluded that the addition of 10%wt CV@NZ is optimal. Both water and oxygen barriers in the case of PLA/TEC/xNZ films were increased as the NZ (%wt.) content increased, and the lowest D_wv_ and Pe_O2_ values were obtained for the PLA/TEC/15NZ film. A decrease in water vapor permeability reported herein is in accordance with the reduction in water vapor permeability by 25%–30% of chitosan/Ag-zeolite films reported recently [77]. In the case of PLA/TEC/xCV@NZ films, water and oxygen barriers decreased as the CV@NZ (%wt.) content increased. The lowest D_wv_ and Pe_O2_ values were recorded for the PLA/TEC/5CV@NZ film. In general, the CV release kinetics for PLA/TEC/xCV@NZ composite films showed that when the amount of CV@NZ (%wt.) increased, the CV release also increased, but the CV release rate decreased. According to the abovementioned results, a medium CV@NZ content (10%wt.) is optimal to obtain a high CV release content and CV control release rate.

Antioxidant and antibacterial activity experiments showed that all PLA/TEC/xCV@NZ films exhibited significant antioxidant and antibacterial activity, in accordance with reports for CV-containing packaging films [79,80,81,82]. In addition, the obtained antioxidant and antibacterial activity of all PLA/TEC/xCV@NZ films had no statistically significant differences in the EC_50_ mean values or the inhibition zones of these films against the Gram-positive *Staphylococcus aureus* and the Gram-negative *Salmonella enterica* ssp. *enterica* serovar Typhimurium pathogens. 

Considering all the parameters tested for PLA/TEC/xCV@NZ active films and assuming the mechanical properties, CV release amount, and CV release rate as the most critical factors for an active packaging film, we conclude that the PLA/TEC/10CV@NZ film is the optimal for a fresh minced pork shelf-life experiment. Both PLA/TEC and PLA/TEC/10CV@NZ films succeeded in extending the shelf life of minced pork meat for two and four days, respectively, compared to minced pork wrapped in commercial paper. The shelf-life extension by four days observed in this study is much higher than the shelf-life extension of pork fillets by two days, recently reported using LDPE/thymol@natural zeolite active films [40]. This result provides a very promising and safe technology for meat preservation because in the case of the PLA/TEC/10CV@NZ film, a small amount of carvacrol encapsulates the active film and releases slowly, protecting the wrapped pork meat. Herein, carvacrol does not need to be used in direct contact with pork meat as it is used in other cases by using the essential oils and their components for meat shelf-life extension [83,84,85]. In addition, the PLA/TEC/10CV@NZ packaging system recorded the lowest and significantly different TBARS values throughout the 10 days of storage time, delaying lipid oxidation deterioration of minced pork. In agreement with the above are the results obtained for the heme iron content determination, where the minced pork wrapped in PLA/TEC/10CV@NZ films recorded higher and significant differently values throughout the storage time. Sensory analysis scores for odor, color, and texture showed that PLA/TEC/10CV@NZ films succeeded in preserving wrapped minced pork in a much better and statistically significant sensory condition than commercial packaging paper. It is noteworthy that minced pork packaged with the PLA/TEC/10CV@NZ film has odor, color, and texture values that are higher than three after 10 days of storage. Last but not least, it is worth mentioning that this novel PLA/TEC/10CV@NZ active film has a self-healable property, which was also observed for the pure PLA/TEC composite matrix recently [22]. As shown in Appendix A, this PLA/TEC/10CV@NZ active film succeeded in completely healing the dog bone-shaped initial scratch after 3 min. 

## 5. Conclusions

Overall, this study provides a novel biodegradable, self-healable active packaging film (PLA/TEC/10CV@NZ) with improved water/oxygen barrier properties and enhanced antioxidant/antibacterial activity, which succeeded in extending the shelf life of fresh minced pork, according to total viable count values, for four days. Simultaneously, this film delays the lipid oxidation, provides higher nutritional values of fresh minced pork, as revealed by the determination of TBARS and heme iron content, and provides a much better and acceptable sensory condition than commercial packaging paper. This film has great potential to be used as an active film to extend the shelf life of fresh meat products.

## Figures and Tables

**Figure 1 antioxidants-13-00776-f001:**
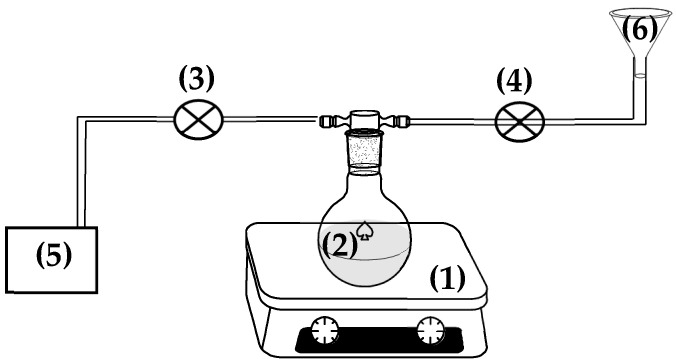
Schematic presentation of handmade apparatus used for the preparation of the CV@NZ nanohybrids: (1) stirrer with a heating plate, (2) spherical glass flask, (3) security valve of the pump, (4) security valve of the CV tank, (5) air vacuum pump, and (6) CV tank. CV: carvacrol and NZ: natural zeolite.

**Figure 2 antioxidants-13-00776-f002:**
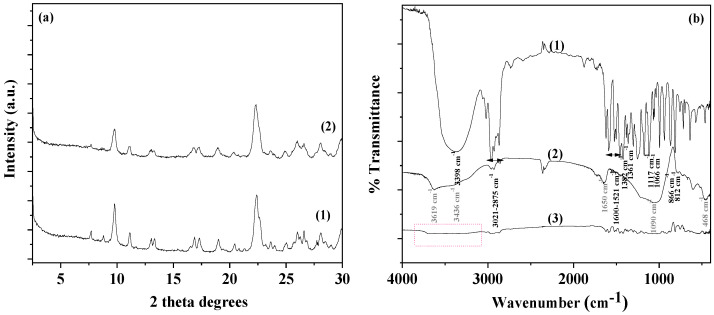
(**a**) XRD plots of (1) pure NZ and (2) the CV@NZ nanohybrid and (**b**) FTIR plots of (1) CV as received, (2) pure NZ, and (3) the CV@NZ nanohybrid. CV: carvacrol and NZ: natural zeolite.

**Figure 3 antioxidants-13-00776-f003:**
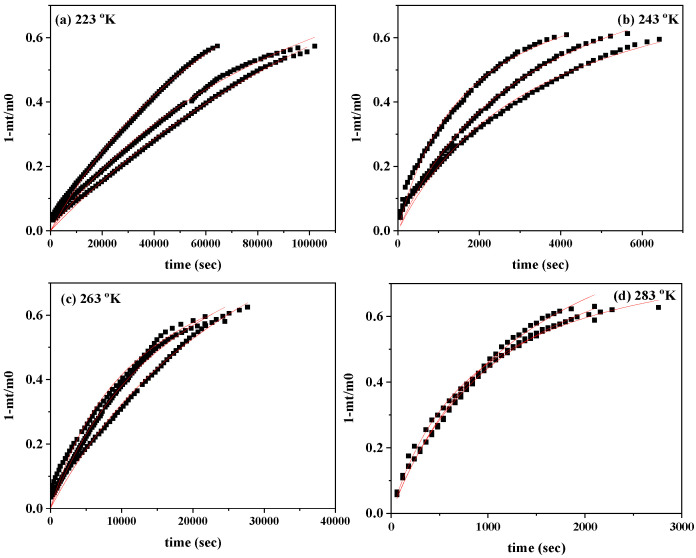
CV desorption isotherm kinetic plots of the CV@NZ nanohybrid at (**a**) 223 °K, (**b**) 243 °K, (**c**) 263 °K, and (**d**) 283 °K. The red lines show the simulation plots according to the pseudo-second order kinetic model. CV: carvacrol and NZ: natural zeolite.

**Figure 4 antioxidants-13-00776-f004:**
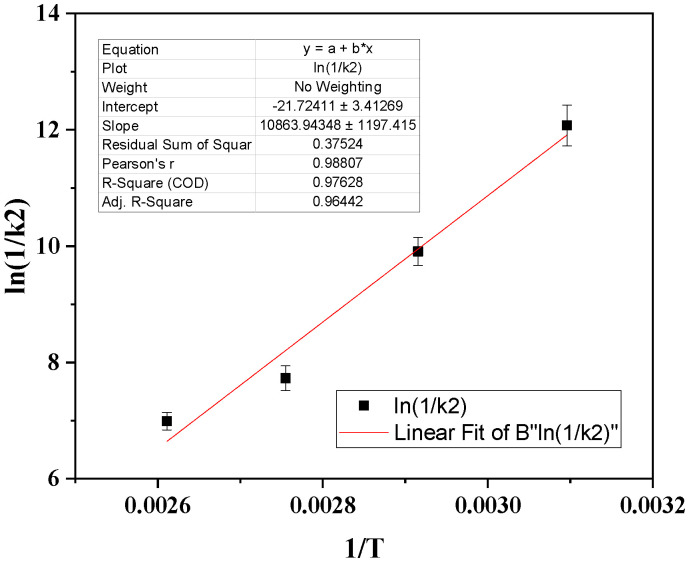
Plot of ln(1/k_2_) values as a function of (1/T).

**Figure 5 antioxidants-13-00776-f005:**
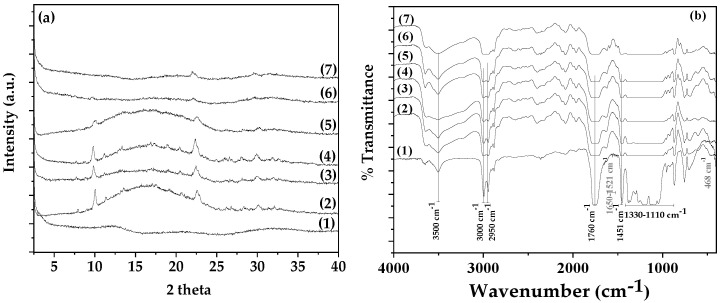
(**a**) XRD and (**b**) FTIR plots of the films: (1) PLA/TEC, (2) PLA/TEC/5NZ, (3) PLA/TEC/10NZ, (4) PLA/TEC/15NZ, (5) PLA/TEC/5CV@NZ, (6) PLA/TEC/10CV@NZ, and (7) PLA/TEC/15CV@NZ. PLA: poly-lactic acid, TEC: triethyl citrate, CV: carvacrol, and NZ: natural zeolite.

**Figure 6 antioxidants-13-00776-f006:**
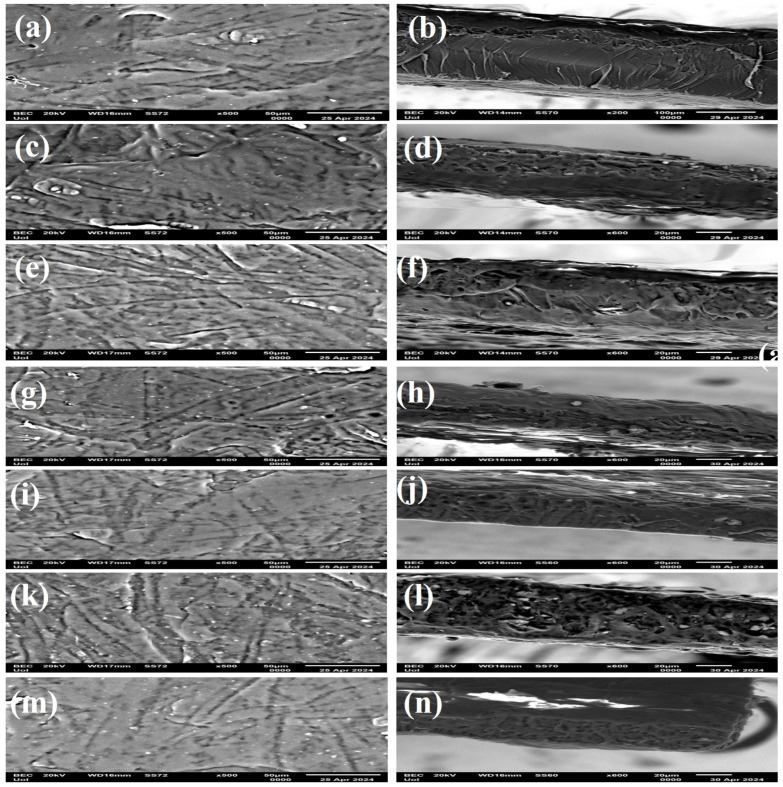
SEM images of the surface (**left column**) and cross-section (**right column**) of the polymer matrix film of PLA/TEC (**a**,**b**) and nanocomposite films of PLA/TEC/5NZ (**c**,**d**), PLA/TEC/5CV@NZ (**e**,**f**), PLA/TEC/10NZ (**g**,**h**), PLA/TEC/10CV@NZ (**i**,**j**), PLA/TEC/15NZ (**k**,**l**), and PLA/TEC/15CV@NZ (**m**,**n**). PLA: poly-lactic acid, TEC: triethyl citrate, CV: carvacrol, and NZ: natural zeolite.

**Figure 7 antioxidants-13-00776-f007:**
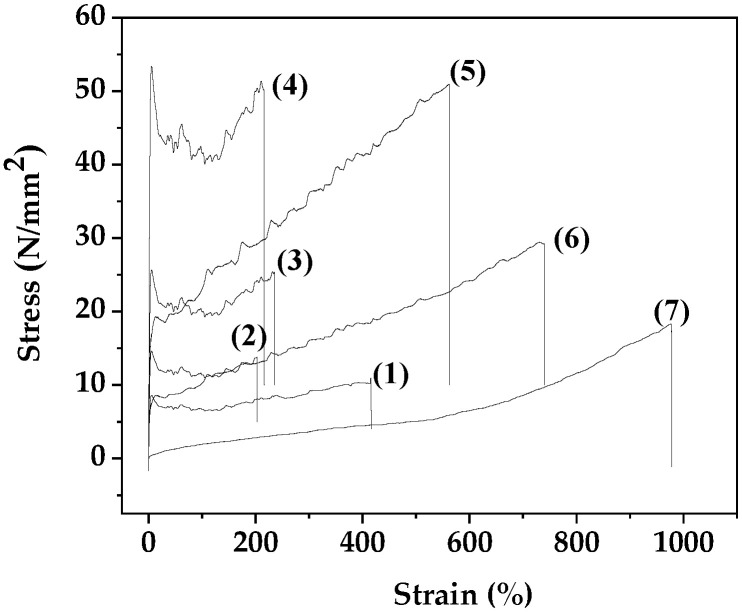
Stress–strain curves of all the obtained PLA/TEC/xNZ and PLA/TEC/xCV@NZ films as well as the pure PLA/TEC film: (1) PLA/TEC, (2) PLA/TEC/5NZ, (3) PLA/TEC/10NZ, (4) PLA/TEC/15NZ, (5) PLA/TEC/5CV@NZ, (6) PLA/TEC/10CV@NZ, and (7) PLA/TEC/15CV@NZ. PLA: poly-lactic acid, TEC: triethyl citrate, CV: carvacrol, and NZ: natural zeolite.

**Figure 8 antioxidants-13-00776-f008:**
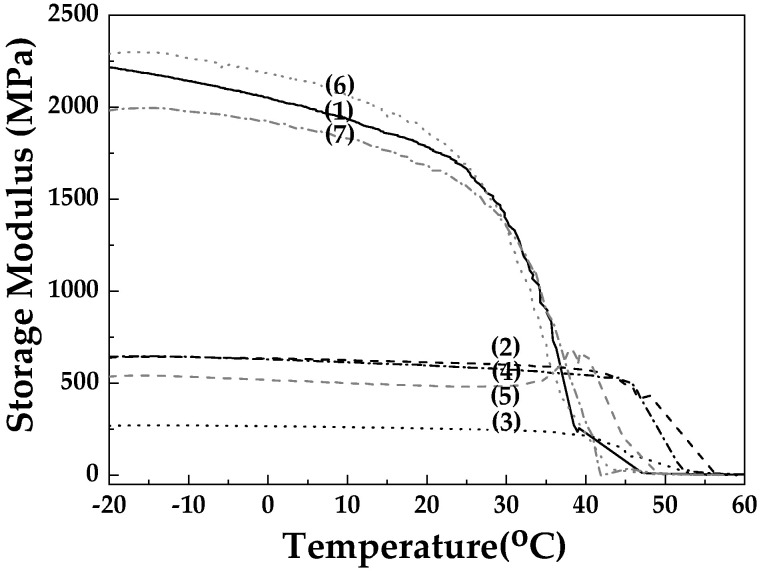
Storage modulus plots as a function of temperature of all the obtained PLA/TEC/xNZ and PLA/TEC/xCV@NZ films as well as the pure PLA/TEC film: (1) PLA/TEC, (2) PLA/TEC/5NZ, (3) PLA/TEC/10NZ, (4) PLA/TEC/15NZ, (5) PLA/TEC/5CV@NZ, (6) PLA/TEC/10CV@NZ, and (7) PLA/TEC/15CV@NZ. PLA: poly-L-lactic acid, TEC: triethyl citrate, CV: carvacrol, and NZ: natural zeolite.

**Figure 9 antioxidants-13-00776-f009:**
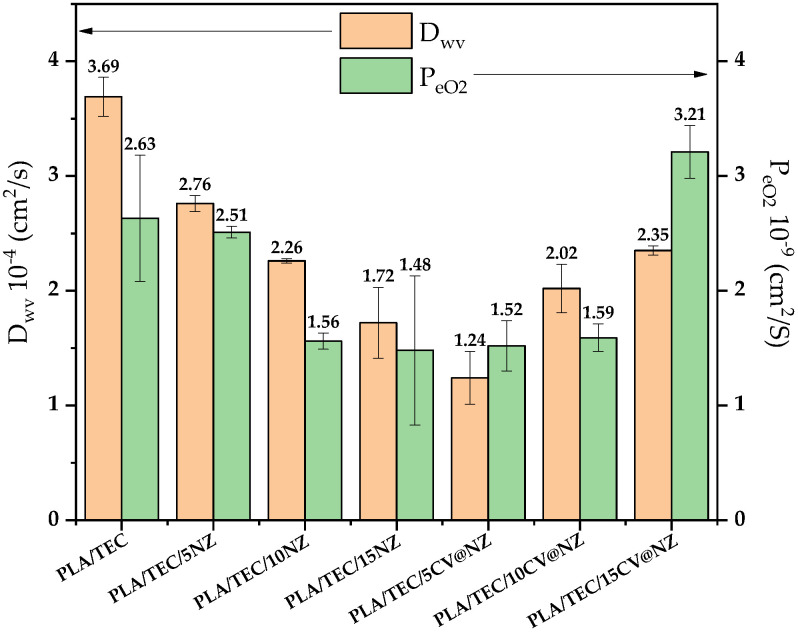
Plot line diagram of mean calculated water vapor diffusion coefficient (D_wv_) values and mean calculated oxygen permeability (P_O2_) values of all the obtained PLA/TEC/xNZ and PLA/TEC/xCV@NZ films as well as the pure PLA/TEC film. PLA: poly-lactic acid, TEC: triethyl citrate, CV: carvacrol, and NZ: natural zeolite.

**Figure 10 antioxidants-13-00776-f010:**
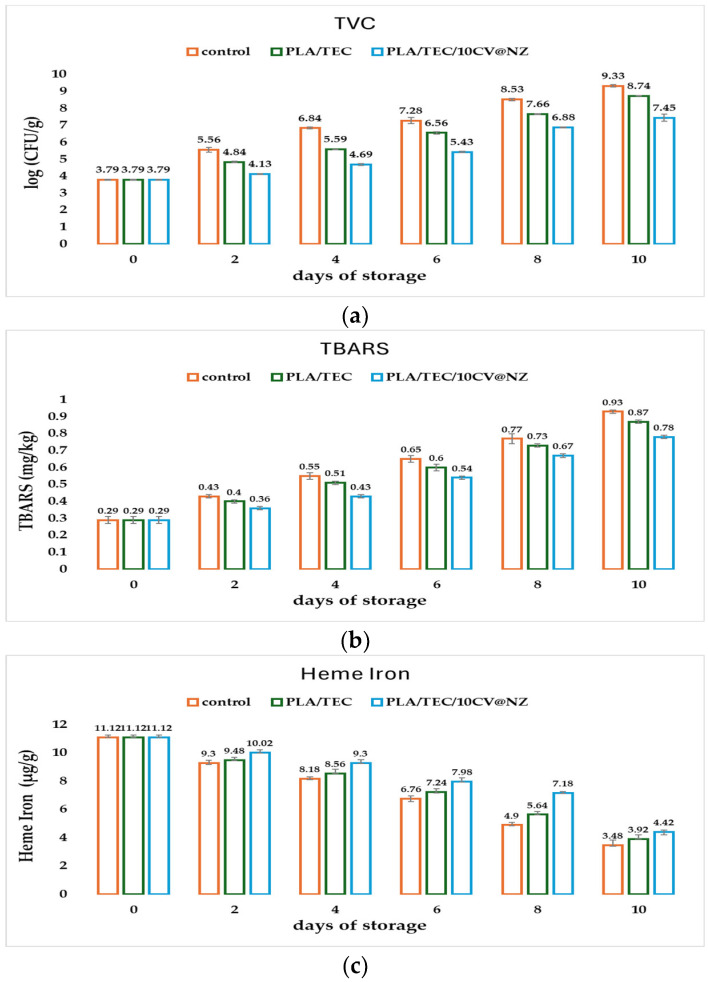
Column bar plots of (**a**) TVC values, (**b**) TBARS values, and (**c**) heme iron content values of minced pork wrapped in the commercial film (control), PLA/TEC film, and PLA/TEC/10CV@NZ film as a function of storage time. PLA: poly-lactic acid, TEC: triethyl citrate, CV: carvacrol, and NZ: natural zeolite.

**Table 1 antioxidants-13-00776-t001:** Sample names, contents of PLA, TEC, pure NZ, and the CV@NZ nanohybrid, and the twin extruder operating conditions (temperature and speed) used for the development of all PLA/TECx composite blends. PLA: poly-L-lactic acid, TEC: triethyl citrate, CV: carvacrol, and NZ: natural zeolite.

Sample Name	PLA (g)	TEC (ml-% *v*/*w*)	NZ (g-% *w*/*w*)	Twin Extruder Operating Conditions		
				T (°C)	Speed (rpm)	Time (min)
PLA/TEC	4	0.6-15	-	180	120	5
PLA/TEC/5NZ	4	0.6-15	0.2-5	180	120	5
PLA/TEC/10NZ	4	0.6-15	0.4-10	180	120	5
PLA/TEC/15NZ	4	0.6-15	0.6-15	180	120	5
PLA/TEC/5CV@NZ	4	0.6-15	0.2-5	180	120	5
PLA/TEC/10CV@NZ	4	0.6-15	0.4-10	180	120	5
PLA/TEC/15CV@NZ	4	0.6-15	0.6-15	180	120	5

**Table 2 antioxidants-13-00776-t002:** Calculated mean post-simulation values for %wt. CV content release, pseudo-second order constant k_2_, desorption capacity at equilibrium q_e_, and R^2^ at 223 °K, 243 °K, 263 °K, and 283 °K. CV: carvacrol.

Temperature	%wt. CV Desorbed	k_2_ (s^−1^)	q_e_ (mg/g)	R^2^
223 °K	56.7 ± 0.6	5.7 × 10^−6^ ± 0.23 × 10^−6^	1.5 ± 0.26	0.994 ± 0.002
243 °K	60.5 ± 1.8	4.98 × 10^−5^ ± 0.31 × 10^−5^	1.16 ± 0.24	0.989 ± 0.008
263 °K	60.7 ± 0.6	4.39 × 10^−4^ ± 0.23 × 10^−4^	0.92 ± 0.07	0.992 ± 0.003
283 °K	61.7 ± 0.23	9.21 × 10^−4^ ± 0.13 × 10^−4^	0.99 ± 0.13	0.995 ± 0.001

**Table 3 antioxidants-13-00776-t003:** Calculated values of the Elastic Modulus (E), the ultimate strength (σ_uts_), and the % elongation at break (%ε) for all the obtained PLA/TEC/xNZ and PLA/TEC/xCV@NZ films as well as the pure PLA/TEC film. PLA: poly-lactic acid, TEC: triethyl citrate, CV: carvacrol, and NZ: natural zeolite.

Sample Name	Ε(MPa)	σ_uts_	%ε
PLA/TEC	564.1 ± 93.7 ^a^	8.0 ± 1.9 ^a^	436.3 ± 285.6 ^a^
PLA/TEC/5NZ	760.33 ± 116.4 ^a^	13.6 ± 1.4 ^b^	202.7 ± 143.8 ^a^
PLA/TEC/10NZ	1536.00 ± 852.3 ^b,a^	24.0 ± 8.7 ^c^	247.8 ± 114.6 ^a^
PLA/TEC/15NZ	2144.25 ± 662.5 ^b^	49.7 ± 4.1 ^d^	216.2 ± 54.1 ^a^
PLA/TEC/5CV@NZ	1093.33 ± 313.7 ^b^	22.5 ± 5.8 ^c^	567.4 ± 252.3 ^b,a^
PLA/TEC/10CV@NZ	404.50 ± 282.1 ^c,a^	10.3 ± 2.1 ^a,b^	724.6 ± 160.0 ^b,a^
PLA/TEC/15CV@NZ	33.00 ± 9.5 ^d^	1.6 ± 0.4 ^e^	993.7 ± 61.5 ^c^

Different letters in each column indicate statistically significant differences at the confidence level *p* < 0.05.

**Table 4 antioxidants-13-00776-t004:** Calculated mean values of the desorption rate (k_2_), the CV desorption equilibrium constant (q_e_), the CV release rate (RR_CV_) as well as the %wt. CV released content (%RC_CV_) for all PLA/TEC/xCV@NZ films. PLA: poly-lactic acid, TEC: triethyl citrate, CV: carvacrol, and NZ: natural zeolite.

	k_2_ (s^−1^)	q_e_ (μg/g)	%RC_cv_	RR_cv_ (μg/s)
PLA/TEC/5CV@NZ	1.65 ± 1.05	0.0168 ± 0.0031	1.74 ± 0.44	0.0082 ± 0.0026
PLA/TEC/10CV@NZ	0.0178 ± 0.0081	0.0471 ± 0.0076	4.20 ± 0.43	0.0049 ± 0.0006
PLA/TEC/15CV@NZ	0.0095 ± 0.0038	0.0611 ± 0.0142	6.17 ± 0.42	0.0039 ± 0.0003

**Table 5 antioxidants-13-00776-t005:** Calculated EC50 mean values for all PLA/TEC/xCV@NZ active films. PLA: poly-lactic acid, TEC: triethyl citrate, CV: carvacrol, and NZ: natural zeolite.

Sample	EC_50_ (mg/L)
PLA/TEC/5CV@NZ	4.65 ± 1.99 ^a^
PLA/TEC/10CV@NZ	2.18 ± 1.15 ^a^
PLA/TEC/15CV@NZ	2.85 ± 2.22 ^a^

Different letters in each column indicate statistically significant differences at the confidence level *p* < 0.05.

**Table 6 antioxidants-13-00776-t006:** Antibacterial activity results of all PLA/TEC/xCV@NZ films against *S. aureus* and *S. Typhimurium.* PLA: poly-lactic acid, TEC: triethyl citrate, CV: carvacrol, and NZ: natural zeolite.

	*S. aureus*		*S. typhimurium*	
Sample	No. of Replicates with Growth in the Contact Area of the Sample	Inhibition Zone Range (mm)	No. of Replicates with Growth in the Contact Area of the Sample	Inhibition Zone Range (mm)
PLA/TEC/5CV@NZ	0/6 ^a^	0.23–0.36 *	0/6 ^a^	0.17–0.24 **
PLA/TEC/10CV@NZ	0/6 ^a^	0.27–0.42 *	0/6 ^a^	0.16–0.21 **
PLA/TEC/15CV@NZ	0/6 ^a^	0.25–0.44 ***	0/6 ^a^	0.18–0.23 *

^a^ 0/6: growth in the contact area in 6 out of 6 replicates, i.e., strong antimicrobial activity. *: the inhibition range refers to 6 out of 6 replicates, **: the inhibition range refers to 5 out of 6 replicates, and ***: the inhibition range refers to 4 out of 6 replicates.

**Table 7 antioxidants-13-00776-t007:** Odor, color, and texture scores of wrapped pork fillets during storage at 4 ± 1 °C. For each packaging treatment, all these properties were ranked from 0 (lowest degree of each characteristic in the tested samples) to 5 (highest degree of each characteristic in the tested samples). PLA: poly-lactic acid, TEC: triethyl citrate, CV: carvacrol, and NZ: natural zeolite.

Sample Name	0 Day	2nd Day	4th Day	6th Day	8th Day	10th Day
**odor**						
CONTROL	5.00 ± 0.00 ^a^	4.45 ± 0.10 ^b^	3.90 ± 0.20 ^d^	3.20 ± 0.10 ^f^	2.85 ± 0.10 ^i^	2.50 ± 0.10 ^m^
PLA/TEC	-	4.55 ± 0.10 ^b^	4.25 ± 0.10 ^e^	3.60 ± 0.20 ^g^	3.25 ± 0.18 ^j^	2.80 ± 0.20 ^l^
PLA/TEC/10 CV@NZ	-	4.70 ± 0.05 ^b.c^	4.50 ± 0.15 ^e^	4.05 ± 0.16 ^h^	3.80 ± 0.10 ^k^	3.20 ± 0.16 ^n^
**color**						
CONTROL	5.00 ± 0.00 ^a^	4.30 ± 0.20 ^b^	3.98 ± 0.29 ^d^	2.85 ± 0.15 ^e^	2.70 ± 0.15 ^h^	2.55 ± 0.12 ^j^
PLA/TEC	-	4.50 ± 0.10 ^b^	4.35 ± 0.27 ^d^	3.45 ± 0.16 ^f^	2.85 ± 0.11 ^h^	2.70 ± 0.16 ^j^
PLA/TEC/10 CV@NZ	-	4.75 ± 0.05 ^c^	4.45 ± 0.27 ^d^	4.02 ± 0.19 ^g^	3.70 ± 0.12 ^i^	3.50 ± 0.15 ^k^
**texture**						
CONTROL	5.00 ± 0.00 ^a^	4.48 ± 0.29 ^b^	3.80 ± 0.20 ^c^	3.32 ± 0.11 ^e^	2.84 ± 0.08 ^g^	2.54 ± 0.10 ^k^
PLA/TEC	-	4.40 ± 0.20 ^b^	4.22 ± 0.15 ^d^	3.70 ± 0.10 ^f^	3.22 ± 0.18 ^h^	2.75 ± 0.10 ^l^
PLA/TEC/10CV@NZ	-	4.56 ± 0.14 ^b^	4.32 ± 0.11 ^d^	3.94 ± 0.18 ^f^	3.60 ± 0.10 ^i^	3.04 ± 0.12 ^m^

Different letters in each column indicate statistically significant differences at the confidence level *p* < 0.05.

## Data Availability

Data are contained within the article and Appendix A.

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
