# Peer review of "Shelf Life of Minced Pork in Vacuum-Adsorbed Carvacrol@Natural Zeolite Nanohybrids and Poly-Lactic Acid/Triethyl Citrate/Carvacrol@Natural Zeolite Self-Healable Active Packaging Films"

_antioxidants, 2024, doi:10.3390/antiox13070776_

Round 1

Reviewer 1 Report

The work on the development of carvacrol@natural zeolite nanohybrid and poly-lactide acid/triethyl citrate/carvacrol@natural zeolite self-healable active packaging films for minced pork shelf-life extension presents an interesting approach to enhancing food preservation using biodegradable materials and natural additives. Here are some comments and questions regarding the study:

1.      The title should be shorter.

2.      The aim should be better described in the abstract and at the end of the introduction part.

3.      Table 2: there are missing some ± sign.

  1. How scalable is the vacuum adsorption desorption method for industrial production? Are there any potential challenges in scaling up the production of CV@NZ nanohybrids?
  2. What is the cost implication of using PLA/TEC/CV@NZ films compared to conventional packaging materials? Is the increased shelf-life and improved food safety enough to justify any additional costs?
  3. How stable are the antimicrobial and mechanical properties of the PLA/TEC/CV@NZ films over an extended period? Is there any degradation in performance over time?
  4. Have there been any considerations regarding regulatory approval for the use of carvacrol and natural zeolite in food packaging? What are the potential regulatory hurdles?
  5. Have sensory tests been conducted with consumers to gauge acceptance of the packaged meat? What were the consumer responses compared to commercially packaged products?
  6. Can the developed nanohybrid and self-healable active packaging films be applied to other types of food products? If so, what adjustments might be necessary for different food matrices?

Overall, this work presents a promising advancement in the field of active packaging, potentially offering significant benefits for food preservation, safety, and sustainability. Further research and development, particularly in terms of scalability, cost-efficiency, and regulatory compliance, will be crucial for its successful commercial implementation.

The work on the development of carvacrol@natural zeolite nanohybrid and poly-lactide acid/triethyl citrate/carvacrol@natural zeolite self-healable active packaging films for minced pork shelf-life extension presents an interesting approach to enhancing food preservation using biodegradable materials and natural additives. Here are some comments and questions regarding the study:

1.      The title should be shorter.

2.      The aim should be better described in the abstract and at the end of the introduction part.

3.      Table 2: there are missing some ± sign.

  1. How scalable is the vacuum adsorption desorption method for industrial production? Are there any potential challenges in scaling up the production of CV@NZ nanohybrids?
  2. What is the cost implication of using PLA/TEC/CV@NZ films compared to conventional packaging materials? Is the increased shelf-life and improved food safety enough to justify any additional costs?
  3. How stable are the antimicrobial and mechanical properties of the PLA/TEC/CV@NZ films over an extended period? Is there any degradation in performance over time?
  4. Have there been any considerations regarding regulatory approval for the use of carvacrol and natural zeolite in food packaging? What are the potential regulatory hurdles?
  5. Have sensory tests been conducted with consumers to gauge acceptance of the packaged meat? What were the consumer responses compared to commercially packaged products?
  6. Can the developed nanohybrid and self-healable active packaging films be applied to other types of food products? If so, what adjustments might be necessary for different food matrices?

Overall, this work presents a promising advancement in the field of active packaging, potentially offering significant benefits for food preservation, safety, and sustainability. Further research and development, particularly in terms of scalability, cost-efficiency, and regulatory compliance, will be crucial for its successful commercial implementation.

Author Response

Authors would like to express their appreciations for Reviewers' effort.

Attached online please find our report.

Regards

The authors

Reviewer 2 Report

Dear Authors,

I have performed the review of your manuscript, I would like to express that your manuscript is well written and presented. I do not have any suggestions for improvement, I am agree with the results and discussions developed in this manuscript.

In my opinion, the manuscrip is using a suitable number of Tables and Figures. Moreover, the authors provide additional information in supplementary materials and non-published document.

This information is very valuable and complement the manuscript submitted. 

Author Response

(The authors gave the same response as above.)

Reviewer 3 Report

The research adds to the body of information on more sustainable packaging materials, but has many deficiencies in the manuscript as noted in the detailed comments. The lack of experimental design and statistical model and improper use of a trained panel to determine acceptability/likeability of the minced pork samples are major reasons to reject the paper from further consideration. Acceptabliity/likeability of samples can only be determined by consumer panel of 50 or more members and not by a panel of 7 experienced members. The research might be better reported and receive more recognition in a packaging journal since much of the study was on the film characteristics and only a minor portion on antioxidation and the minced pork properties.

Line(s)          Comment

2-4                “Minced pork shelf-life in carvacrol@natural zeolite nanohybrid by vacuum absorption and poly-lactide 2 acid / triethyl citrate / carvacrol@natural zeolite self-healable active pack-3 aging films.”

16                 The phrase “in the spirit of the bioeconomy” is not understood.

28-29           The actual antioxidant and antibacterial data could be given.

38-41           Reference(s) needed to substantiate that these statements are factual.

43-47           Reference(s) needed to substantiate that these statements are factual.

61-65           Reference(s) needed to substantiate that these statements are factual.

72-73           Reference(s) needed to substantiate that these statements are factual.

113               The muscles, grind size, and fat content for the fresh minced pork should be given.,

129               Each figure and table must be able to be interpreted independently from the text so acronyms and abbreviations such as CV and NZ must be defined.

161-162       The variables in one or both equations must be defined.

182               See comment for l. 129.

188               The model and manufacturer of the hydraulic press with heated platens must be given.

201               The model and manufacturer of the hydraulic press must be given.

209, 215      “were tested, and” “Tensed” usually refers to a human body movement.

245-247       The evidence that this moisture analyzer test will release carvacrol similarly to thymol must be stated since the OH group is at different positions on the phenol ring between the two compounds.

260               The model and manufacturer of the pH probe and meter must be given.

306               The source and composition of the commercial paper used by the Aifantis meat company must be given.

308-310       It must be described if the commercial packaging was opaque or transparent and if the refrigerated storage was in the dark or light.

309               The model and manufacturer of the refrigerator must be given.

332               The materials and method of filtering must be described.

349-352       The sensory evaluation method is invalid. If the 7 experienced members were considered a trained panel, then only a consumer panel and not a trained panel can determine like or acceptability of food samples. If the 7 experienced members are considered a consumer panel, then a statistically valid explanation for the use of less than 50 to 100 consumer panelists must be provided to justify the small number of panelists. With a trained panel, the training procedures, number and types of samples for training, and number of training sessions must be given and for both trained and consumer panels, the selection of panelists, environmental conditions and facilities for testing, number of samples per testing session, and number of testing sessions must be given.

354-358       The number of replications of the experiment must be given.

363               The experimental design and statistical model for SPSS analyses must be given.

384-385       Each figure and table must be able to be interpreted independently from the text so acronyms and abbreviations must be defined or described.

390-391       This is repetitive of l. 384-485.

392-400       The usefulness and/or relevance of this information is not clear.

425               Each figure and table must be able to be interpreted independently from the text so acronyms and abbreviations must be defined or described.

429               Each figure and table must be able to be interpreted independently from the text so acronyms and abbreviations must be defined or described.

440-442       This speculation since the desorption and the availability of the natural zeolite absorption active sites were not directly measured.

461-466       Reference(s) needed to substantiate that these statements are factual.

486-487       Each figure and table must be able to be interpreted independently from the text so acronyms and abbreviations must be defined or described.

516-518       Each figure and table must be able to be interpreted independently from the text so acronyms and abbreviations must be defined or described.

552-556       Each figure and table must be able to be interpreted independently from the text so acronyms and abbreviations must be defined or described.

579               It would be clearer to intermix dotted, dashed, and solid lines in addition to the color to differentiate the treatment lines or to number them as in Fig. 5 and 7.

580-581       Each figure and table must be able to be interpreted independently from the text so acronyms and abbreviations must be defined or described.

599-600       Each figure and table must be able to be interpreted independently from the text so acronyms and abbreviations must be defined or described.

624               Each figure and table must be able to be interpreted independently from the text so acronyms and abbreviations must be defined or described.

640               Each figure and table must be able to be interpreted independently from the text so acronyms and abbreviations must be defined or described.

653-655       Each figure and table must be able to be interpreted independently from the text so acronyms and abbreviations must be defined or described.

693-695       Each figure and table must be able to be interpreted independently from the text so acronyms and abbreviations must be defined or described.

744               Reference(s) needed to substantiate that heme iron content is an important nutritional index for pork and other meats.

777               The scale used for the scores must be defined.

777-778       If only the means within each column are compared, then only letters a-c would be used. If all 16 means were compared to determine any interactions between the treatment and storage time, then the footnote in l. 778 should indicated “Different letters within a variable indicated differences (P<0.05).”

778               Use of “statistically significant” and the probability level in the same sentence is redundant.

790-852       Most of this section is a repeat of the results with minimal discussion of the reasons for differences and importance of the results and almost no comparison of the results with other studies.

854-860       This repeats the results and does not provide useful information on the applicability, importance or commercialization of the results.

862               If the supplemental information is important to providing or explaining necessary results, then it should be imbedded in the manuscript text instead of being presented as supplemental information.

904-1057     Consistency is needed for the journal names as some are given as the full name and other journal names are abbreviated.

970               No journal name or access information is given.

992-997       English-speaking journals or an English translation of the journal article title are required.

Author Response

(The authors gave the same response as above.)

Round 2

Reviewer 3 Report

It is appreciated that the authors attempted to address the manuscript deficiencies. The confusion of experiment sequence with experimental statistical design and the lack of an explanation on how the descriptive panel determined the important sample characteristics indicate that recognized scientific standards were not followed. Having the same data in a figure and in a table is repetitive and wastes valuable journal space.

Line(s)          Comment

16-17           This sentence is not appropriate as the research does not investigate the environmental friendliness of the packaging.

17                 Delete “Following the spirit of the circular economy” as this phrase has no specific meaning.

100               It must be clarified if the pork was from the front leg or hind leg as these have different muscle fiber types and color.

340-343       A trained sensory panel cannot determine the degree of likeability or acceptability by any method. A descriptive analysis panel determines the important product characteristics and then evaluates the degree of each characteristic in the tested samples.

345-355       The experimental design (completely randomized, block, nested, factorial) was not described and the model used to statistically analyze the treatments, including replication as a variable, was not given.

450               This is repetitive of l. 435.

779-780       The scale used for the scores must be defined.

950, 981, 983 There are no authors for the citations.

Author Response

Dear Reviewer

First we would like to thank you for your effort to improve our work

Please find attached our response to your suggestions/comments.

Sincerely yours

Prof. Dr. Constantinos Salmas

Prof. Dr. Aris Giannakas
